# Optimized Design of Touching Parts of Soil Disinfection Machine Based on Strain Sensing and Discrete Element Simulation

**DOI:** 10.3390/s23146369

**Published:** 2023-07-13

**Authors:** Jianmin Gao, Yuhao Shen, Benlei Ma

**Affiliations:** School of Agricultural Equipment Engineering, Jiangsu University, Zhenjiang 212013, China

**Keywords:** vertical rotary tillage, variable disinfection, STM32, discrete element, EDEM, simulation analysis

## Abstract

With the increasing level in the intensification of agricultural production in China, continuous cropping obstacles have become a problem that needs to be solved. The use of vertical rotary tillage technology and soil disinfection technology is an effective solution. In this paper, a vertical rotary soil-tilling variable disinfection combine was developed and an on-board control system with STM32 as the control core was designed to realize the real-time acquisition of powder monopoly torque information and the variable application of soil disinfection chemicals. Based on the obtained experimental soil parameters, a discrete element soil particle model was established, and orthogonal experiments were conducted to analyze the single-blade roller tillage process, and the optimal operating parameters were finally selected as 500 mm powder monopoly depth, 320 r/min knife roller speed, and 0.26 m/s forward speed, respectively. The field experiment found that the average tillage depth of the implement was 489 mm, the stability coefficient of tillage depth was 94.50%, the uniformity coefficient of soil disinfection was 85.57%, and the applied amount and the speed ratio coefficient of the given flow were linearly related, respectively. This research provides a technical reference for the deep tillage and soil disinfection of the powder monopoly.

## 1. Introduction

Continuous cropping obstacles refer to the abnormal growth and development of crops caused by the continuous cultivation of the same crop or related crops on the same soil, which leads to the formation of a special soil environment and the breeding of a large number of harmful bacteria. As the scale of agricultural production has expanded and the level of intensification has increased in recent years, the problem of continuous crop disorder has become increasingly prominent, which has seriously restricted the sustainable development of the agriculture [1,2]. Since the 1980s, China’s vegetable farming industry has developed rapidly. The area of protected vegetable fields has reached 4.8 million hectares, accounting for 30% of the vegetable area, and 3.7% of the national cultivated area, respectively [3]. However, because of continuous cropping for many years, protected vegetable fields have had to endure the breeding and spread of pests and diseases, and the content of the soil organic matter has decreased as a result. Long-term shallow rotary tillage cannot be used to solve the problem of continuous cropping, and, instead, a hard plow bottom layer is formed under the crop root system, and the land becomes less permeable to pests. Furthermore, the air permeability of the land becomes poor, the ability to store water and moisture is reduced, the vegetable growth and development is poor, and the efficiency is significantly reduced [4,5,6,7].

Soil disinfection combined with tillage is an effective solution to continuous cropping obstacles, which can be used to kill most of the virus bacteria in the soil at the same time as cultivation. At present, the vertical spiral blade roll has been widely used in afforestation and reforestation operations due to its reliability and ability to crush soil quickly [8,9]. The principle of vertical rotary tillage is precisely based on the use of the mechanical vertical spiral blade roll to grind and suspend the soil naturally, integrating the functions of plowing, harrowing, crushing, and loosening the soil. This technology can be used to improve the soil structure and porosity, keep the soil relatively loose for a long time, break the bottom of the plow, increase the depth of the soil tilth, and improve the physical and chemical properties of the soil [10]. At the same time, after vertical rotary tillage, the soil environment is conducive to the infiltration of disinfectants and full mixing with the soil. Therefore, developing agricultural machinery with a simple structure and stable operation is important, which integrates vertical rotary tillage and the soil variable disinfection technique.

The Guangxi Academy of Agricultural Sciences put forward the concept of vertical rotary tillage for the first time in 2008, and described the tools and methods that can be used for vertical rotary tillage [11]. This technology uses vertical spiral blade rolls to complete deep cultivation, and perform high-speed rotary cutting and soil preparation, and leads to the characteristics of an undisturbed soil layer, flat land, and loose soil after tillage [12]. Yang et al. analyzed the operating performance factors for a vertical rotary tillage blade based on the SPH method in 2019, and established a mathematical model for the structural parameters and power consumption of the blade [13]. It was found that the range of the power consumption is relatively small when using a small blade roll diameter, a small pitch, and a small number of blade combinations. Xiao et al. proposed a double spiral vertical rotary tillage blade tool structure in 2020 [14], which can decrease the force amplitude and torque amplitude by 78% and 52.8%, respectively. However, field experiments were not conducted, and the double spiral structure may have the problem of ‘too fast soil rising’.

Soil disinfection technology has attracted the attention of international scholars since the 1980s [15,16]. At present, the most commonly used soil disinfection technologies include solar disinfection, steam sterilization, chemical disinfection, and hot water disinfection [17]. Bourbos et al. developed a solar soil disinfection method in 1993. However, the disinfection cycle for this technology is long, and it cannot be used in continuous cloudy and rainy weather, and cannot achieve uniform disinfection for deep soil [16]. Nishi et al. and Kita et al. changed the physical and chemical properties of the soil through a hot water disinfection device, but their machines were not able to uniformly disinfect the deep soil layer and had certain requirements for the quality of the steam [18,19,20,21]. Gay et al. mounted a steam generator and diversion device onto a tracked power machine in 2010 to kill pathogens in the soil through circulation of steam at high temperatures [22,23]. Peruzzi et al. developed a self-propelled soil disinfection machine [24], which sprayed exothermic active substances, and stirred the soil at the same time as rotary tillage. It was found that when the forward speed of the machines is kept at 0.08–0.15 km/h, the soil diseases and pests at a depth of 200 mm can be effectively controlled. Jinok et al. proposed steam disinfection technology [25,26], which uses the atomization principle to diffuse high-temperature gases into the soil, and also uses sensor technology to measure and adjust the distance between the spray rod and the soil in real time to improve the uniformity of disinfection. Fanari et al. proposed a soil disinfection scheme based on microwaves that are irradiated from an antenna and is used to kill weed pests and heat up the soil [27], resulting in a non-linear, multi-physical medium heating phenomenon. Xu et al. designed a Spike-hood soil steam processor [28], but this machine is generally used in sandy loam soil, and is not suitable for other soil types. Liu et al. developed a combined operation machine for split deep rotary tillage and soil disinfection [29]. The pesticide and soil were fully mixed by spraying onto the surface and inside of the soil. The cultivation disinfection depth reached 35 cm, the stability coefficient of the cultivation depth was 94.84%, and the disinfection uniformity coefficient reached 64.23%, respectively. Fang et al. designed a solid fumigant disinfector, but their disinfectant could not enter the deeper layers of the soil [30]. Ma et al. used microwave heating of soil to verify its disinfection effect, but no field trials were conducted [31].

Based on the above research, it can be inferred that the current soil disinfection machines cannot be used to disinfect and sterilize deep bacteria, and disinfection and sterilization are uneven, while vertical rotary tillage technology can crush the soil deeply, which can promote the penetration of disinfectants and improve the mixing degree of the chemicals and soil, thereby producing a deep uniform disinfection and sterilization. The purpose of this paper was to establish a discrete element soil particle model using EDEM 2018 software, simulate and analyze the process of single blade roll tillage, investigate the influence of different operating parameters on power consumption, and design an on-board control system with STM32 as the control core to achieve the real-time collection of information, such as dust monopoly torque and the variable application of soil disinfection chemicals. The tillage and disinfection performance of the machine was verified through experiments. The objective of this paper was to develop a machine to solve the continuous crop barrier, improve the crop growth environment, and achieve green harvest.

## 2. Structure Design

### 2.1. Whole Machine Structure

The structure of the machine designed in this paper is shown in Figure 1, which includes a vertical rotary tillage system and a variable disinfection system. The frame was welded onto the rear of a gearbox. The vertical rotary tillage system is mainly composed of a transmission system and a spiral blade roll. The tillage depth can reach more than 40 cm. The variable disinfection system mainly includes an electric throttle valve, a flow control box, a disinfection spray bar, etc., which together coordinate the operation to achieve a variable disinfection of the soil.

The machine is connected to the front power machinery through a three-point suspension device, and the angle of the blade into the soil was adjusted by controlling the length of the hydraulic top link. The power output shaft drives the three spur gears located in the same plane behind it through a pair of bevel gears to engage and rotate. The lower side of the spur gear is connected to a spiral blade through a flange plate to realize the vertical rotary tillage of the soil on the silt ridge. The variable disinfection system is mounted onto a rack behind the gearbox. After the spiral blade has been buried, the disinfectant in the disinfection kit can be sprayed onto the soil surface through a filter, reflux pump, and throttle valve in turn. A schematic diagram for soil disinfection is shown in Figure 2.

### 2.2. Design of the Blade Roll Torque Detection System and the Variable Disinfection System

In this study, a variable disinfection system was designed with STM32-M3 as the core, including an upper computer and two lower computers, equipped with a UCOSIII real-time operating system and an EmWin graphical user interface [32]. STC8F2K08S2 was used as the core of the first lower computer to measure the soil cutting torque of the spiral blade roll, while the voltage signal was measured using a full bridge circuit connected by resistive strain gauges, and the torque data were transmitted to the upper computer via a DL-20 wireless communication module. A second lower computer with STM32 as the core was installed in a flow control box, which was equipped with a Hall speed sensor, relay, stepper motor controller, and Hall flow meter, and communication was established with the upper computer through a DL-20 wireless module to complete the real-time regulation and flow control. STM32F103-V3 was used as the core of the upper computer, carrying an EmWin graphical application interface on an LCD screen for the real-time display of tension, flow, speed, and positioning information obtained from a GPS module collected by the lower computer, while the torque information for the machine was visualized with Graph control. In addition, the operator can use a SD card to import the prescription information and continuously extract the disinfection information from the prescription map based on the GPS latitude and longitude and send the control signal to the second lower computer through a wireless communication mode to realize the variable disinfection operation. This system was also used to set the normal operation flow threshold for an abnormal flow alarm. The overall scheme and system interface of this system are shown in Figure 3 and Figure 4, respectively.

## 3. Design of the Simulation System

### 3.1. Soil Sample Acquisition

To establish the soil model and ensure the accuracy of the simulation, the soil in the test plot was sampled and then analyzed to obtain the relevant physical properties of the soil, and the characteristic parameters are given in Table 1.

All the soil samples were obtained from the field trial area in Pei County, Xuzhou City, Jiangsu Province, China, and these soil samples were measured using the five-point sampling method. Since this study involved a deep tillage machine, the samples were taken from the surface to 60 cm below it in three layers: upper (0–20 cm), middle (20–40 cm) and lower (40–60 cm), respectively.

#### 3.1.1. Shear Test for the Soil

Samples were obtained from the test field inspection area according to the upper, middle, and lower layers, and four soil samples were taken for each different soil depth. A preloading instrument was used to conduct an 8 h preloading test for the soil samples to ensure that the compactness of the soil samples were the same. The permeable stone was placed into a shear box and then pushed into the soil sample to ensure that its shape was not damaged, followed by the permeable stone and pressure transfer plate and the adjustment of the pressurized frame. A vertical stress of 100, 200, 300, and 400 kPa was applied to each soil layer, the shear rate was set to 0.8 mm/min, and the load was applied uniformly, respectively. The start time at which the force expression starts to change was recorded until either the soil sample was damaged, or the maximum deformation displacement reached 6 mm, and the process of force expression with the displacement change was recorded. The shear strength curves for each soil layer were plotted according to the data collected, as shown in Figure 5.

The maximum shear stress for each group of tests was determined using the least squares method to fit the line of the shear strength versus the vertical pressure according to the Coulomb formula, and the shear strength parameters for the different soil layers were finally determined, as shown in Table 2.The internal friction angle is expressed as the slope of the fitted line, and the soil cohesion values displayed in Table 2 is shown as the intersection of the fitted line with the vertical axis.

#### 3.1.2. Soil Strength Test

The prepared soil sample was placed onto the compression plate of the strain-controlled, unconfined compression apparatus, following which the swing wheel was rotated, and the position of the upper compression plate was adjusted to make light contact with the upper surface of the soil sample. The wheel was rotated to keep the strain rate between 1% and 3% per minute, respectively, and the values measured using the dynamometer and the displacement meter were recorded at any time until the values measured by the dynamometer reached a peak value or became constant. Then, a 3% to 5% axial strain was applied to complete this group of tests. The readings obtained from the dynamometer were then used to calculate the pressure for the axial stress using Equation (1):(1)$$\sigma =\frac{CR}{A_{\alpha }}$$
where σ is axial stress, Kpa; C is the dynamometer rate coefficient, N/0.01 mm; R is the dynamometer value, 0.01 mm; and $A_{\alpha }$ is the corrected sample area, cm^2^, respectively.

The relationship curve between the axial stress and strain was drawn according to the test calculation results, as shown in Figure 6.

### 3.2. Soil Particle Modeling

The soil particle shape was equated to a sphere with the radius of 8 mm, and Poisson’s ratio was chosen to be 0.35 based on the soil (loam) texture and the empirical equation. The basic parameters for the soil model obtained from the literature and the soil test data are given in Table 3 [29,33,34,35].

The interaction parameters for the soil particles include the static friction coefficient, the rolling friction coefficient, and the recovery coefficient. The determination of this part of the parameters required a large number of experiments to be carried out, and this paper used the soil accumulation angle as the base parameter to determine this part of the interaction parameters. The particle parameters that are similar to those obtained from the soil accumulation angle test were selected in the GMEE material library of EDEM 2018 software, and a virtual simulation model was established to simulate the falling process for the soil in its natural state as an approximate representation of the particle interaction parameters. After several rounds of simulation tests, the interaction parameters for each material were finally determined, as shown in Table 4.

According to the above tests, the soil showed a loam texture in the test area, and the soil moisture content in this area was deemed to be high. Therefore, the Hertz-Mindlin with bonding model was selected as the soil particle contact model. In the discrete element software, the model for the bonding bond interaction between these soils includes the following five parameters: normal stiffness, tangential stiffness, critical positive stress, critical shear stress, and bond radius, respectively while the process model for soil particles from the bonded state to failure damage is expressed as follows [36,37]:(2)$$\begin{array}{c} \delta F_{n}=-V_{n}S_{n}At\\ \delta F_{t}=-V_{t}S_{t}At\\ \delta T_{n}=-\omega_{n}S_{t}Jt\\ \delta T_{t}=-\omega_{t}S_{n}\frac{J}{2}t\\ A=\pi {R_{1}}^{2}\\ J=\frac{1}{2}\pi {R_{1}}^{4} \end{array}$$
where $F_{n}$ and $F_{t}$ are the bonding bond normal force, tangential force, N; $T_{n}$ and $T_{t}$ are the bonding bond normal torque, tangential torque, N·m; $V_{n}$ and $V_{t}$ designate the normal velocity, tangential velocity, m/s; $\omega_{n}$ and $\omega_{t}$ are the normal angular velocity, tangential angular velocity, rad/s; $S_{n}$ and $S_{t}$ represent the normal stiffness, tangential stiffness, N/m^3^; *A* is the contact area, m^2^; *J* is the moment of inertia, m^4^; $R_{1}$ isthe bonding radius, m; and *T* is the unit time step, s.

Bonding bond breaking condition [36,37]:(3)$$\begin{array}{c} \sigma_{max}<-\frac{F_{n}}{A}+\frac{2T_{t}}{J}R_{1}\\ \tau_{max}<-\frac{F_{t}}{A}+\frac{2T_{n}}{J}R_{1} \end{array}$$
where $\sigma_{max}$—critical normal stress, Pa; and $\tau_{max}$—critical tangential stress, Pa.

The bond radius between the soil particles can be calculated from the soil density, according to Equation (4). In addition, the normal stiffness and the critical positive stress were obtained using the unconfined compressive strength test. The critical positive stress was defined as the maximum stress that the soil can withstand in the vertical direction, i.e., σ, when the small principal stress is 0, which was calculated according to Equation (3). Normal stiffness refers to the ability of the vertical direction to resist elastic deformation when subjected to a force and is related to the bond radius of the particles in the discrete element, which was determined using the ratio of the maximum stress to the bond radius in this paper. The critical shear stress refers to the critical positive stress in the soil direct shear test, which is the point at which the soil Coulomb equation is fitted in the unconfined compressive strength test. The tangential stiffness is expressed by the ratio of the maximum shear stress to the bond radius. In summary, the parameters of the Hertz-Mindlin with bonding contact model are shown in Table 5.
(4)$$\begin{array}{c} w_{t}=\frac{\rho_{s}V_{s}}{\rho_{w}V_{w}}\times 100\%\\ V_{w}=\frac{4}{3}\pi R_{w}^{3}\\ V_{s}=\frac{4}{3}\pi R_{1}^{3}-\frac{4}{3}\pi R_{w}^{3} \end{array}$$
where $\rho_{s}$—density of water, $g/{cm}^{3}$; $\rho_{w}$—density of the soil, $g/{cm}^{3}$; $V_{s}$—volume of soil particle water layer, ${cm}^{3}$; $V_{w}$—volume of soil particle, ${cm}^{3}$; and $R_{w}$—radius of a single soil particle, cm, respectively.

### 3.3. Simulation and Postprocessing

According to the operating depth and width of the spiral single blade roll, the soil bin was established, the blade roll model was imported, and the motion parameters of the spiral blade roll were set. The whole simulation process is shown in Figure 7.

The simulation shows that the spiral blade roll will form pits in the ground when cutting the soil, and the tilled soil will be scattered in a circular shape across the soil surface with a diameter of approximately 230 mm. As the blade roll moves forward, a large number of soil particles are thrown backwards, filling up the holes again, and as the soil is in a loose state after tillage, and the broken bonding bonds are not recreated, this means that the soil layer is “elevated” as a whole after tillage. Figure 8a shows the cross-sectional view of the soil after plowing, which also confirms that vertical rotary tillage does not break the original soil structure, and helps to maintain the physical and chemical properties of the soil. Figure 8b shows the bounding bond breakage with a constant increment in the number of normal operating orders broken with time, indicating the uniformity of the tillage.

In this paper, the ratio of the broken bonds to the total bonds in the tillage area was used to represent the crushing rate, and the EDEM postprocessing module ‘Set Bin Group’ was used to establish the tillage data collection area and divide the collection grid. The bonding bonds formed and broken between all the particles in the collection area were calculated to yield a crushing rate of 92.34%.

The power consumption is an important parameter to measure the performance of the machine [38]. The power consumption of vertical rotary tillage mainly includes three processes: transmission power consumption, forward power consumption, and soil cutting power consumption, and this paper took the spiral single blade roll cutting simulation as the research object to assess the influence of the forward speed, blade roll speed, and tillage depth on the operating power consumption. In the EDEM simulation, analysis of the above two types of power consumption was reflected in the cutting torque and the forward direction resistance suffered by the blade roll. In the postprocessing interface, the total torque applied to the blade roll in the z-direction, and the total resistance applied to the blade roll in the x-direction can be respectively used to generate line graphs according to time steps, as shown in Figure 9 and Figure 10, respectively.

The above figure shows that in the time period of 0~2 s particle bed generation, the torque and forward resistance are 0; in the time period of 2~4 s, the spiral blade rolls into the soil, the torque increases rapidly, and the tension swings back and forth between ±500 N; and in the time period of 4~10 s, the normal operation time for the blade roll, the torque gradually stabilizes over a certain interval, while the tension swings rapidly to the most significant value point due to the beginning of the forward movement, and remains stable over this time interval.

### 3.4. Validation of the Simulation System

To ensure the accuracy of the simulation results, the torque was assessed using the resistance-strain type measurement method, while the torque value was calculated using the changing resistance value, and the calculated results were compared with the simulation results to verify whether this simulation system is reliable. The actual operating condition was integrated, and a position 30 mm below the reinforced rib plate was selected as the patching site. The small axis calibration method was used to calibrate the pasted strain gauge torque sensor. By hanging weights onto the end of the arm bar, a torque was generated on the blade roll, and the strain voltage output from the bridge was measured using an oscilloscope. The calibration results are shown in Figure 11.

The torsion measuring device was fixed with tape, and the data were transmitted to a PC via a serial port. A single blade roll was installed onto the machine, and the torsion measuring device was then added. The simulation operation process was simulated using the motion parameters at the time of the simulation to test the single blade roll in the field, and the operation process is shown in Figure 12.

To cope with the sudden influx of data with positive and negative oscillations and increase the data processing time, the reference voltage was set at 12 mV, the expected bit value was 768, and the actual bit value was at 771, respectively. The measured voltage signal bit value graph is shown in Figure 13. The actual working state bit average value was 2381, and the difference was 1610, corresponding to a voltage signal of 25.15625 mV and a torque of 157.41, respectively. The simulated torque value was 148.02 N × m, and the error value was 5.97%, respectively. The reason for this analysis was that the field conditions are more complex, and the weeds and debris among the soil and rocks can produce errors in the experimental results. In summary, these experimental results are basically consistent with the simulation results, thereby indicating that the simulation model established in this study is feasible.

## 4. Experiments and Analysis

### 4.1. Virtual Simulation Experiment

The orthogonal test is an effective method for constructing multi-factor and multi-level tests [39]. From the perspective of practical simulation research, the forward speed, the blade roll speed, and the tillage depth were selected as the test factors, and the power consumption for soil cutting and the forward power consumption were taken as the test indices. According to the pre-simulation test and the vertical rotary tillage requirements, with the actual operation standards, the test factor level table was set, as shown in Table 6.

In this paper, a single blade roll was removed to simulate the operation process and the magnitude of the power consumption of the computer tool. This virtual simulation test was used to mainly study the forward power consumption and the soil cutting power consumption of the spiral blade roll in the operating state of vertical rotary tillage, and the soil cutting power consumption was calculated according to Equation (5):(5)$$P_{z}=\frac{T_{z}n_{1}}{9550}$$
where $P_{z}$—cutting power consumption, kW; $T_{z}$—cutting torque, N·m; and $n_{1}$—blade roll speed, r/min, respectively.

Forward power consumption is the rate of energy consumed by the machine to keep moving in the forward direction, and was calculated using Equation (6):(6)$$P_{x}=F_{x}V_{x}$$
where $P_{x}$—forward power consumption, W; $F_{x}$—rotary tillage pull, N; and $V_{x}$—forward speed, m/s, respectively.

The three-factor, three-level orthogonal virtual test was designed according to Table 6 [40,41], where A represents the forward speed; B represents the tillage depth; and C represents the blade roll speed, respectively. The test results are shown in Table 7.

It can be observed from the analysis table that within the range of the set test group, the total power consumption for single blade roll vertical rotary tillage basically ranges between 2500–6500 W, while the vertical rotary soil-tilling variable disinfection combine designed in this paper is composed of three blade rolls, and its rotary tillage power consumption varies within 20 kW. According to the above analysis, the proportion of the forward power consumption to the total power consumption of the ridge was 4.5~12.5%, which is basically consistent with the proportion of the power consumption for the traditional rotary cultivators.

#### 4.1.1. Analysis of the Range

To analyze the role of each factor on the power consumption, the power consumption for soil cutting along with the forward power consumption was determined with range analysis through the R value obtained from the size of the analysis of the influence in the order of priority, and the range analysis table is shown in Table 8. According to the range analysis for the two types of power consumption, the factors that affect the power consumption performance of the machine follow the order of tillage depth, forward speed, and blade roll speed, respectively.

#### 4.1.2. Analysis of Variance

Most of the power consumption for the tillage implements arise from the soil cutting, so soil cutting power consumption is the focus of the research in the study of tillage power consumption. To further investigate the influence of each operating parameter on the power consumption for soil cutting, a three-factor ANOVA was conducted for the power consumption of soil cutting using SPSS 22.0 software with the three factors of forward speed, tillage depth, and blade roll speed as variables, and Table 9 shows the ANOVA table obtained for the three-factor test.

The ANOVA results are consistent with the results of the ANORA. The *p* values for the forward speed and tillage depth are 0.029 and 0.012, respectively, which indicates that the forward speed and the tillage depth have significant effects on the power consumption. The power consumption for soil cutting, the operation quality, the blade roll operation intensity, and the agronomic requirements for vertical rotary tillage were integrated; the selected tillage depth was 50 cm, the forward speed was 0.26 m/s, and the blade roll speed was not determined.

#### 4.1.3. Single-Factor Test for the Blade Roll Speed

To investigate the effects of the blade roll speed on the power consumption for soil cutting, single-factor tests were designed for analysis. Five sets of single-factor tests were designed with a fixed forward speed of 0.26 m/s and a tillage depth of 500 mm in the speed range of 260–340 r/min, respectively. The experimental data were calculated to derive the average soil cutting power consumption and the torque variance. The changes in the soil cutting power consumption and the operational stability with a change in the blade roll speed were analyzed, and the results are shown in Figure 14.

The analysis shows that the power consumption for soil cutting rises slowly with increasing rotational speed, but the trend was not determined to be significant, which verifies the accuracy of the above orthogonal test. The torque variance reflects the stability of the machine operation and the impact load problem. The smaller the variance, the more stable the operation of the implement, and the smaller the impact load on it. Therefore, the final combination of optimal tillage parameters was determined to be 500 mm tillage depth, 0.26 m/s forward speed, and 320 r/min blade roll speed, respectively.

### 4.2. Field Experiment

#### 4.2.1. Tillage Performance Field Test

For a vertical rotary tillage machine, the tillage depth and the tillage stability are the most important parameters. The length of the stable working area was set to 12 m, the working depth was measured by using a steel ruler, and 29 sample points were taken from the stable working area in a serpentine way to measure the tillage depth. The actual point-taking diagram is shown in Figure 15.

The average tillage depth, the standard deviation of the tillage depth, the coefficient of variation of the tillage depth, and the stability coefficient of the tillage depth for the machine were all calculated using the following equations:(7)$$\bar{S}=\frac{\sum_{i=1}^{N}S_{i}}{N}$$
where $\bar{S}$ is the average tillage depth, mm; $S_{i}$ is the tillage depth of the ith sample, mm; and N represents the sample size, 29, respectively.

Equation (8) was used to determine the standard deviation in the tillage depth.
(8)$$\sigma =\sqrt{\frac{\sum_{i=1}^{N}{\left(S_{i}-\bar{S}\right)}^{2}}{N-1}}$$

Equation (9) was used to determine the coefficient of variation for the tillage depth.
(9)$$V=\frac{\sigma }{\bar{S}}\times 100\%$$

Equation (10) was used to determine the stability coefficient for the tillage depth.
(10)U=(100−V)×100%

The actual tillage depth measurement results are shown in Table 10. The measured tillage depth sample point data were used to calculate the average tillage depth $\bar{S}$ = 489.07 mm, the standard deviation of the tillage depth σ = 26.92, the coefficient of variation of tillage depth V = 5.50, and the stability coefficient of the tillage depth U=94.50%, respectively.

#### 4.2.2. Variable Disinfection Field Test

Eight hundred grams of high potassium water-soluble fertilizer was dissolved in 25 L of water and mixed well as a disinfection agent for spraying, and the variable disinfection effect was assessed by detecting the content and distribution of potassium elements in the soil before and after machine operation. Before the operation, the soil area to be tested was sampled according to the serpentine sampling points to detect the amount of potassium originally contained in the soil to be sprayed, and the samples were taken according to the surface soil and subsoil, with the surface soil obtained directly from the ground, and the subsoil sample taken at a distance of 15 cm below the ground surface. A flow rate ratio of four was set, and the test area was then operated on. The soil samples were resampled after 12 h of resting to form a control with the previous sampling points, and the potassium content of the soil samples was measured using a soil nutrient quick tester after the sampling was completed. The potassium content for each point was obtained using a soil nutrient tester, and the results are shown in Figure 16.

Analysis of the actual potassium application to the soil shows that the potassium water-soluble fertilizer can easily penetrate into the lower soil layer. The uniformity coefficient and the application amount distribution stability coefficient were calculated for the application amount of the disinfecting agent to the surface layer of the soil, as according to Equations (11)–(13).
(11)$$\sigma =\sqrt{\frac{\sum_{i=1}^{N}{\left(X_{i}-\bar{X}\right)}^{2}}{N-1}}$$
(12)$$V=\frac{\sigma }{\bar{X}}\times 100$$
(13)$$U=\left(100-V\right)\times 100\%$$

Substituting the actual application rates for the above topsoil into the above equation results in a standard deviation of 15.99 for the distribution of applied potassium in the topsoil, a coefficient of variation of distribution *V* = 14.43, and a coefficient for the stable nature of the distribution of the application rate of *U* = 85.57%, respectively.

To further evaluate the accuracy of the spraying flow rate, the flow rate was adjusted to a flow rate ratio of 3 and 5, respectively. The above process was repeated, and only the amount of potassium applied to the topsoil was assessed; the results are shown in Table 11.

The average value for the flow rate ratio was extracted, and the three sets of test data with coefficients of three, four, and five were removed. The data point with a flow rate ratio of four was used as the basis to draw the theoretical application line, and the positions of the points with flow rate ratios of three and five were observed, as shown in Figure 17.

## 5. Discussion

In this research, we aimed to develop a soil disinfection machine and assess its rationality by the machine’s tillage performance and the disinfection effect observed in the field. The results of the simulation and field trial showed that the machine could achieve a tillage depth stability factor of 94.50% and a disinfection uniformity factor of 85.57%, respectively. Compared with the vertical spiral blade roll used in this research, Zhang and Liu et al. used multiple pairs of square knives as rototiller rolls to tillage the soil [42], and their tillage depth stability reached 95.24% and 94.84%, respectively, which were close to the results of this research. However, since this type of blade roll throws soil in all directions, which can affect the mixing of the disinfection chemicals with the soil, and with their disinfection uniformity coefficient being 64.23%, it can be shown that the vertical spiral blade roll is more suitable for vertical tillage as well as soil disinfection.

## 6. Conclusions

In this paper, a vertical rotary tilling-soil variable disinfection combine was designed, simulated, and tested. The study shows that this machine is capable of vertical rotary tillage operation while achieving the variable chemical disinfection agent spraying of the soil. After determining the soil parameters, an accurate discrete element soil particle model was established. A spiral blade roll-soil interaction model was established based on the discrete element method, and the accuracy of the prototype was verified through applying field tests. The results show that the relative error between the actual soil cutting torque and the simulation results under the same motion parameters was 5.97%. The simulation system was also used to conduct orthogonal tests, and the optimal operating parameters were finally selected from the viewpoint of reducing the operating power consumption and improving the operating effect: tillage depth of 500 mm, blade roll speed of 320 r/min, and a forward speed of 0.26 m/s, respectively. A blade roll torque detection system and a variable disinfection system based on STM32 was designed to achieve a variable disinfection operation for the soil through the synergistic cooperation of the GPS and a prescription map with an electronically controlled throttle valve as the main actuating device. The test results show that the uniformity coefficient for disinfection in the upper soil layer reached 85.57%, the disinfection agent can penetrate to a distance of 15 cm below the surface, and the application amount and given flow rate ratio coefficients are basically linear.

In this study, the design of the spiral blade roll was only based on the empirical formula and the existing research base. The parameters, such as the spiral lift angle of the blade roll have an important influence on the power consumption of the machine, as well as the operation quality. In future research, we will use discrete element software to perform soil cutting simulations for different parameters of the spiral knife rolls, including variable pitch, segmented spiral, and double spiral blade rolls to achieve further optimization of the implements. In addition, the variable disinfection method used in this study does not form a closed-loop control, although a subsequent information collection module such as a flow meter was added. In future research, we will establish a PID closed-loop control model to correct the actual output flow rate by imposing a corrective control relationship using the flowmeter signal as feedback information.

## Figures and Tables

**Figure 1 sensors-23-06369-f001:**
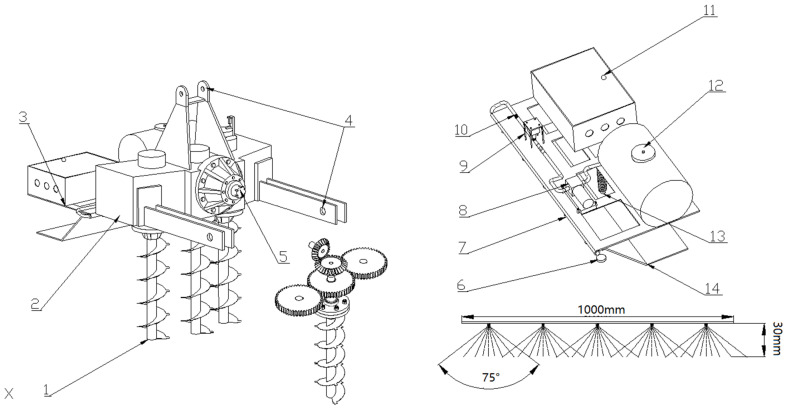
Structure diagram for a vertical rotary tilling-variable soil disinfection combine.The indicated values are as follows: (1) vertical spiral blade roll; (2) gearbox; (3) disinfection frame; (4) three-point hitch; (5) power input shaft; (6) pressure gauge; (7) disinfection spray bar; (8) reflux pump; (9) throttle valve; (10) flowmeter; (11) flow control box; (12) disinfection kit; (13) mulch spring; and (14) mulch plate.

**Figure 2 sensors-23-06369-f002:**
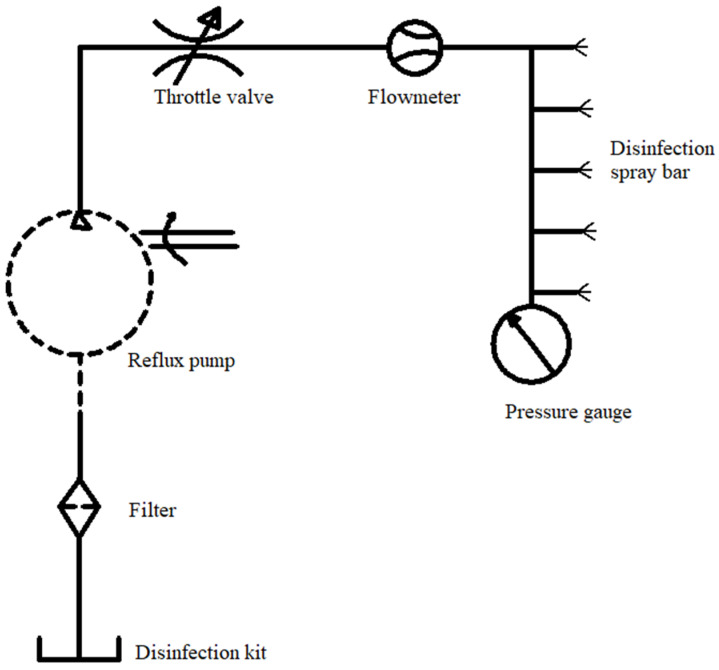
Schematic diagram for soil disinfection.

**Figure 3 sensors-23-06369-f003:**
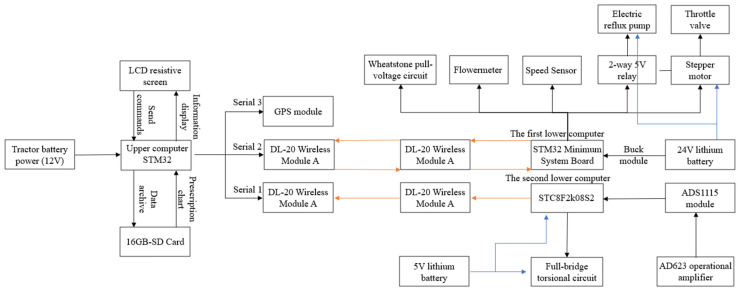
System architecture diagram.

**Figure 4 sensors-23-06369-f004:**
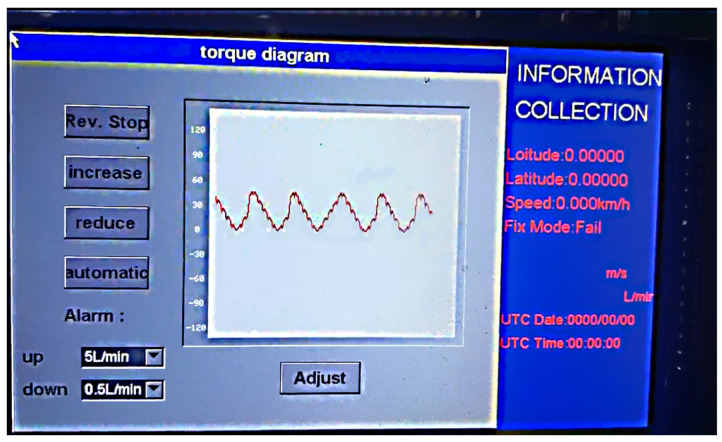
System interface.

**Figure 5 sensors-23-06369-f005:**
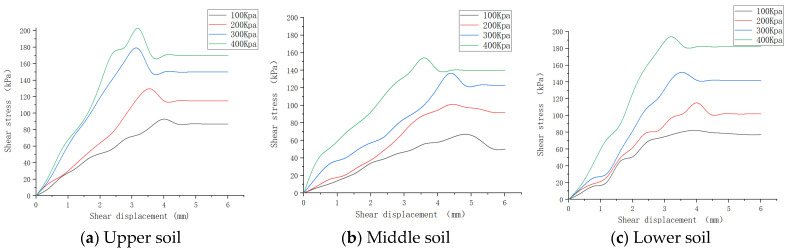
Soil shear strength curves.

**Figure 6 sensors-23-06369-f006:**
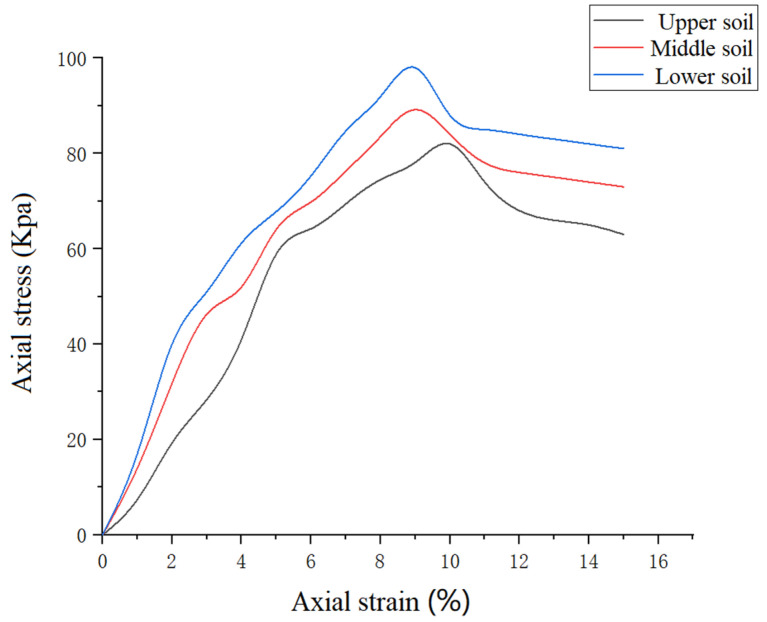
Relationship curve between the axial stress and strain.

**Figure 7 sensors-23-06369-f007:**
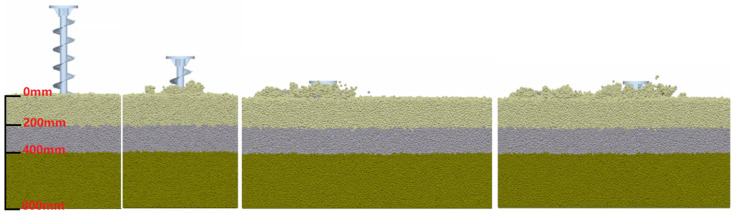
The vertical rotary tillage process.

**Figure 8 sensors-23-06369-f008:**
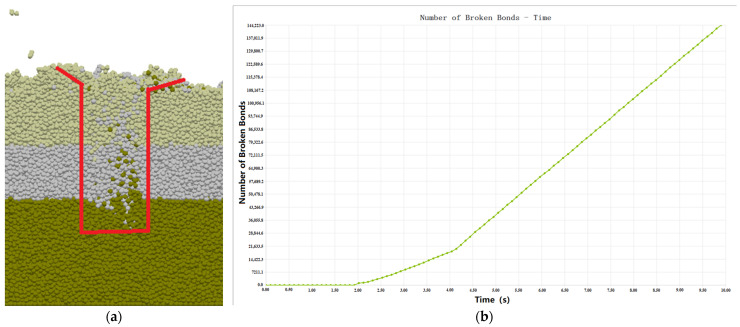
Analysis of the effects of vertical rotary tillage. (**a**) Section diagram of the soil after tillage; and (**b**) division of the bounding keys.

**Figure 9 sensors-23-06369-f009:**
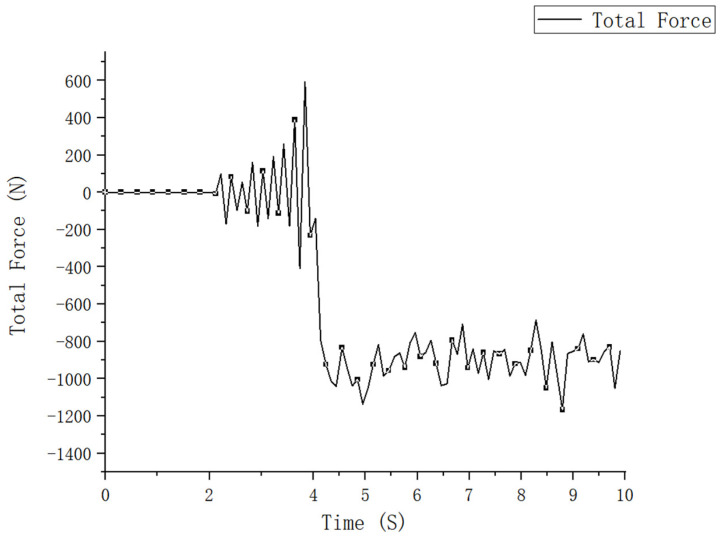
Forward resistance.

**Figure 10 sensors-23-06369-f010:**
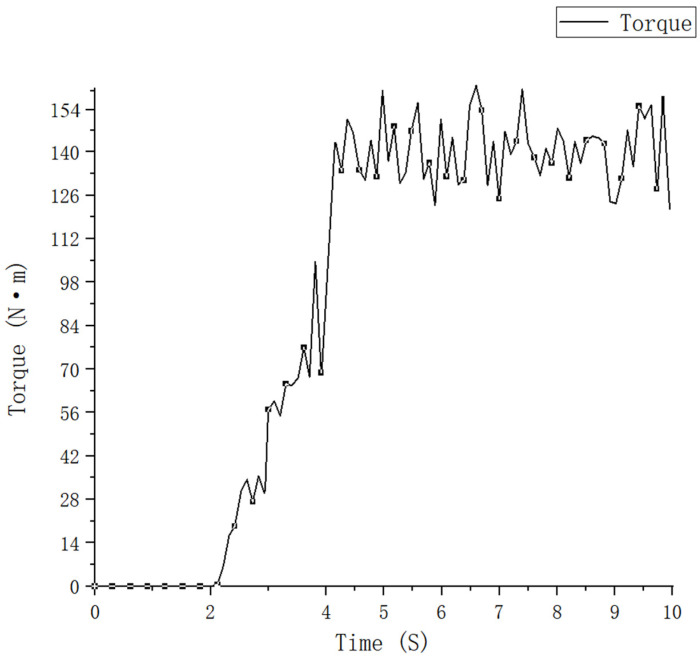
Cutting torque.

**Figure 11 sensors-23-06369-f011:**
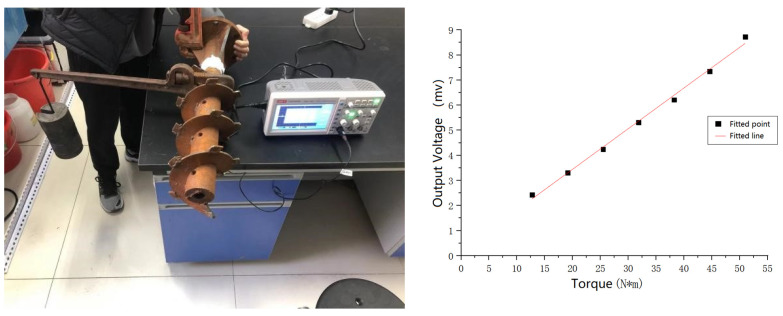
Calibration process and fitting result.

**Figure 12 sensors-23-06369-f012:**
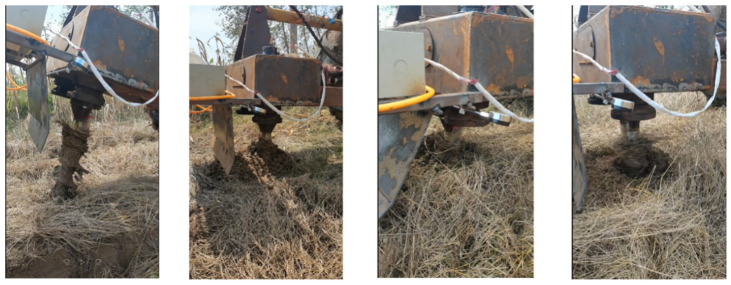
Single blade roll operation process.

**Figure 13 sensors-23-06369-f013:**
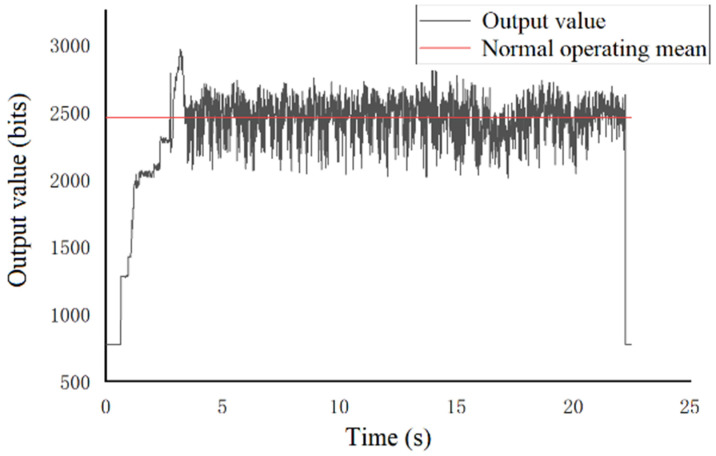
Torque output bit value.

**Figure 14 sensors-23-06369-f014:**
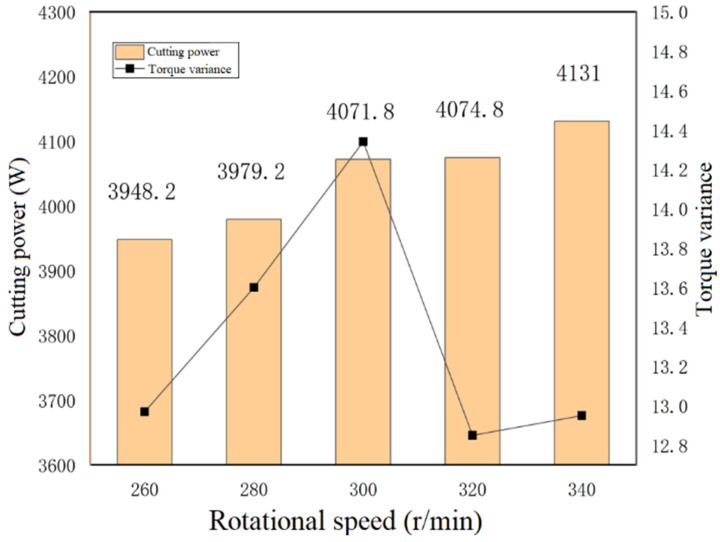
Power consumption and torque variance for soil cutting at the same speed.

**Figure 15 sensors-23-06369-f015:**
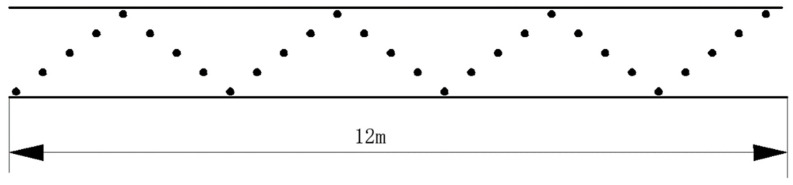
Diagram of the tillage depth measurement points.

**Figure 16 sensors-23-06369-f016:**
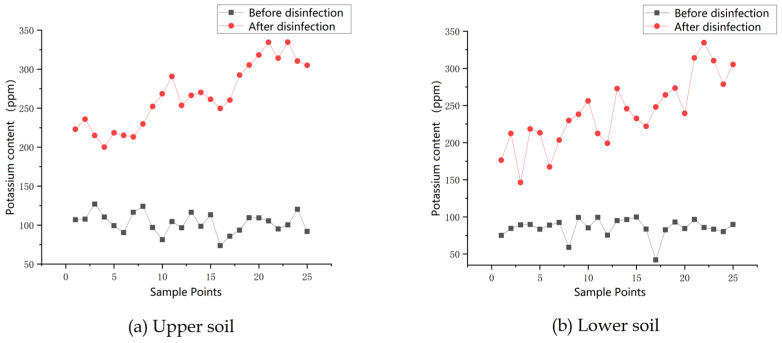
Comparison of the potassium content in the soil before and after the application machine operation.

**Figure 17 sensors-23-06369-f017:**
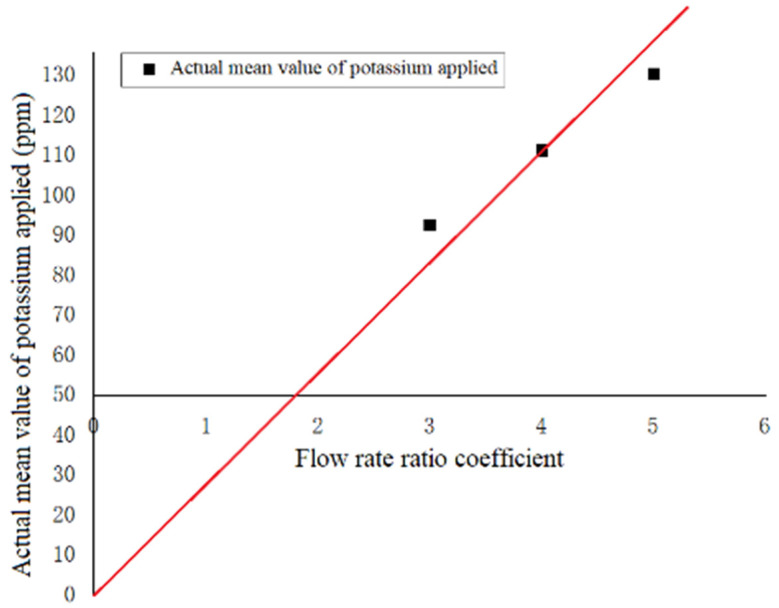
Comparisons in potassium application.

**Table 1 sensors-23-06369-t001:** Characteristics of the assessed soil.

Depth	Cone Index	Moisture Content	Volume Field Capacity	Soil Density	Particle Size Distribution	Soil Compaction	Consistency (Standard Deviation)
Upper soil (0~200 mm)	0.40 MPa	34.52%	32.1%	2.45 g cm^−3^	11.49% clay, 41.13% silt, and 47.38% sand, respectively	0~24.7 N/cm^3^	Plastic limit = 24.75 ± 0.14%, liquid limit = 50.31 ± 0.17%
Middle soil (200~400 mm)	0.28 MPa	32.78%				24.7~38.4 N/cm^3^	
Lower soil (400~600 mm)	0.20 MPa	30.65%				40 N/cm^3^	

**Table 2 sensors-23-06369-t002:** Soil shear strength parameters.

Layer	Internal Friction Angle/°	Soil Cohesion/kPa
Upper soil	16.55	40.5
Middle soil	20.37	42.8
Lower soil	20.75	56.1

**Table 3 sensors-23-06369-t003:** Basic parameters for the soil simulation.

	Poisson’s Ratio	Density/kg/m^3^	Shear Modulus/pa	Particle Radius/mm
Upper soil	0.35	1732	1.1 × 10^6^	8
Middle soil	0.35	1844	1.5 × 10^6^	8
Lower soil	0.35	1893	1.7 × 10^6^	8

**Table 4 sensors-23-06369-t004:** Interaction parameters for the soil particles.

	Recovery Coefficient	Static Friction Coefficient	Rolling Friction Coefficient
Upper soil	0.4	0.6	0.15
Middle soil	0.45	0.6	0.18
Lower soil	0.5	0.6	0.2
Between different soil layers	0.5	0.6	0.2

**Table 5 sensors-23-06369-t005:** Hertz-Mindlin with bonding contact model parameters.

Bonding Contact Model	Value
Normal stiffness/N/m^3^	9.6 × 10^6^~1.05 × 10^7^
Tangential stiffness/N/m^3^	4.74 × 10^6^~6.38 × 10^6^
Critical positive stress/Pa	8.2 × 10^4^~9.2 × 10^4^
Critical shear stress/Pa	4.05 × 10^4^~5.61 × 10^4^
Bonding radius/mm	8.54~8.79

**Table 6 sensors-23-06369-t006:** Test factor levels.

Levels	Forward Speed/m/s	Tillage Depth/mm	Blade Roll Speed/r/min
1	0.2	400	260
2	0.26	500	300
3	0.32	600	340

**Table 7 sensors-23-06369-t007:** Test results.

	Test Factors			Test Index		
No.	A	B	C	Cutting Power Consumption P/W	Forward Power Consumption P/W	Total Power Consumption P/W
1	1	1	1	2318.22	161	2479.22
2	1	2	2	3240.94	181.3	3422.24
3	1	3	3	4296.46	201.2	4497.66
4	2	1	3	2987.02	194.96	3181.98
5	2	2	1	3948.19	396.77	4344.96
6	2	3	2	5147.43	416.23	5563.66
7	3	1	2	3400.21	381.22	3781.43
8	3	2	3	4873.57	427.52	5301.09
9	3	3	1	5641.32	805.37	6446.69

**Table 8 sensors-23-06369-t008:** Analysis of range table.

Soil Cutting Power Consumption	Test Index		
	Forward Speed	Tillage Depth	Blade Roll Speed
K1	9855.62	8705.45	11,907.73
K2	12,082.64	12,062.7	11,788.58
K3	13,915.1	15,085.21	12,157.05
R	1353.16	2126.59	122.82
Forward power consumption			
K1	543.50	737.18	1363.14
K2	1007.96	1005.59	978.75
K3	1614.11	1422.80	823.68

**Table 9 sensors-23-06369-t009:** Three-factor ANOVA (analysis of variance).

	Sum of Squares of Deviations	df	Mean Square Error	F	P
Forward speed	2,755,211.73	2	1,377,605.87	32.91	0.029 *
Tillage depth	6,789,781.32	2	3,394,890.66	81.12	0.012 *
Blade roll speed	23,569.70	2	11,784.85	0.28	0.78
Sum of residuals	83,704.86	2	41,852.43		
	9,652,267.62	8			
R^2^: 0.991					

* *p* < 0.05 means significant effect.

**Table 10 sensors-23-06369-t010:** Tillage depth test results.

Sample Points	Tillage Depth/mm	Sample Points	Tillage Depth/mm
1	512	16	445
2	503	17	527
3	462	18	505
4	453	19	505
5	512	20	486
6	519	21	519
7	508	22	479
8	452	23	486
9	474	24	495
10	427	25	464
11	457	26	485
12	508	27	460
13	510	28	523
14	489	29	502
15	516		

**Table 11 sensors-23-06369-t011:** Potassium application at different rate ratios.

Flow Rate Ratio of 3		Flow Rate Ratio of 5	
**Sample Points**	Potassium Application/ppm	Sample Points	Potassium Application/ppm
1	108.07	1	140.87
2	76.96	2	145.31
3	101.09	3	111.4
4	77.73	4	118.05
5	119.6	5	150.97
6	83.23	6	148.46
7	107.58	7	150.14
8	89.5	8	120.48
9	105.54	9	114.38
10	90.77	10	121.3
11	94.83	11	129.97
12	76.76	12	144.71
13	112.34	13	144.46
14	78.69	14	122.07
15	106.83	15	111.06
16	72.61	16	116.19
17	77.27	17	146.69
18	81.76	18	120.9
19	101.51	19	132.63
20	98.75	20	145.87
21	103.59	21	108.48
22	88.53	22	114.43
23	89.48	23	114.65
24	85.28	24	120.89
25	76.09	25	153.39
Average value	92.18		129.91

## Data Availability

All datasets used in this study are included in the manuscript.

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
