# Peer review of "Optimized Design of Touching Parts of Soil Disinfection Machine Based on Strain Sensing and Discrete Element Simulation"

_sensors, 2023, doi:10.3390/s23146369_

Round 1

Reviewer 1 Report

I have a few remarks: 

Could you change in the work (figure, table and text) "kPa" to "kPa".

Check the presented numbering of equations.

Since you are using the EDEM software, could you show the particle motion describing the force-displacement equations (normal and tangential).

Author Response

File Name:     Review Report of Research Article sensors

Tittle:     Optmized design of touching parts of soil disinfection machine based on strain sensing and discrete element simulation

Subject:        Authors response to the Editor and Reviewer

Respected Editor,

                       We would like to thank the Editor and Referee for reviewing and providing many useful suggestions to make our manuscript more effective. We revised the manuscript and tried our best by incorporating the suggestions of reviewers properly. In this response, we are presenting a point-by-point summary of the reviewer's Points.

Responses to reviewer 1.

Major concerns:

  1. Could you change in the work (figure, table and text) "kPa" to "kPa".

Response: Thank you for your valuable Point. We have standardized the units in the paper and changed “Kpa” to “kPa”.

  1. Check the presented numbering of equations.

Response: We have checked the numbering of the equations and rearranged the numbering order.

  1. Since you are using the EDEM software, could you show the particle motion describing the force-displacement equations (normal and tangential).

Response: We chose the Hertz-Mindin model for the selection of the particle contact model, and the mechanical properties between the bonding bonds were used to model the soil properties, the following literature was referenced (We have added the corresponding reference at the model equation for the bonding bond):

[1]. Jiménez-Herrera, N., G.K. Barrios, and L.M. Tavares, Comparison of breakage models in DEM in simulating impact on particle beds. Advanced Powder Technology, 2018. 29(3): p. 692-706. doi: https://doi.org/10.1016/j.apt.2017.12.006

[2]. Zhang, X., et al., Calibration and Verification of Bonding Parameters of Banana Straw Simulation Model Based on Discrete Element Method. Transactions of the Chinese Society for Agricultural Machinery, 2023. 54(05): p. 121-130. doi: 10.6041/j.issn.1000-1298.2023.05.012

The purpose of this research using EDEM is to simulate the resistance and torque in the soil when the blade roller is working to obtain the operating parameters of the machine, while the force and displacement between the particles is not the main subject of this work. We will discuss the force, displacement and distribution of the particles in detail when we research the state of the soil after tillage in the future.

Reviewer 2 Report

Dear Authors,

I have finished my review on your paper. I think that it is a good one but still it needs some improvements in the format, information given, references and organization of the text.

The Introduction should benefit from a broader international literature on the topic of tillage. Searching on MDPI I have found these papers on the topic:

https://www.mdpi.com/1424-8220/21/18/6288

https://www.mdpi.com/1999-4907/9/11/665

The text seems to be very wordy and less organized. The methods should be more concise and al the relevant parts of the text should be moved in the corresponding section. Format should be also carefully checked and more emphasis should be put on the sensor-based part of the study.

The text should be carefully checked for typos or for uncomplete information. For example there are references provided in the form Author et. al. without the year in () or a typical reference in [].

Best regards and good luck,

Rev.

English needs minor improvements.

Author Response

File Name: Review Report of Research Article sensors

Tittle:     Optmized design of touching parts of soil disinfection machine based on strain sensing and discrete element simulation

Subject:        Authors response to the Editor and Reviewer

Respected Editor,

               We would like to thank the Editor and Referee for reviewing and providing many useful suggestions to make our manuscript more effective. We revised the manuscript and tried our best by incorporating the suggestions of reviewers properly. In this response, we are presenting a point-by-point summary of the reviewer's Points.

Responses to reviewer 2.

Major concerns:

  1. The Introduction should benefit from a broader international literature on the topic of tillage. Searching on MDPI I have found these papers on the topic:

https://www.mdpi.com/1424-8220/21/18/6288

https://www.mdpi.com/1999-4907/9/11/665

Response: Thank you for your valuable Point. We have included in the second paragraph of the introduction a description of the international application of vertical spiral blade roll to support the hypothesis of this research.

  1. The text seems to be very wordy and less organized. The methods should be more concise and al the relevant parts of the text should be moved in the corresponding section. Format should be also carefully checked and more emphasis should be put on the sensor-based part of the study.

Response: We examined the entire presentation and formatting of the paper and made improvements to make it more readable.

  1. The text should be carefully checked for typos or for uncomplete information. For example there are references provided in the form Author et. al. without the year in () or a typical reference in [].

Response: We have systematically checked the entire paper, adjusted the position and format of citations in the text for references, and added missing parts.

Reviewer 3 Report

The manuscript was written in a very harmonious way between the paragraphs.

The introduction prepares for reading and sets out the scenario in which the research was developed.

The topic of soil disinfection by mechanical means is very useful and topical and the exhibited machine can have a positive effect on development.

The survey was conducted in a very comprehensive manner and a very timely and well explained methodology was applied.

The results have been clearly expressed and the discussion is consistent with the results obtained.

I suggest only one note to improve the manuscript:

In table 1 the authors should clarify in the expression of consistency whether the range has been added or whether it is the standard deviation or the standard error.

For everything else, I congratulate the authors and invite them to continue this research.

Author Response

File Name: Review Report of Research Article sensors

Tittle:     Optmized design of touching parts of soil disinfection machine based on strain sensing and discrete element simulation

Subject:        Authors response to the Editor and Reviewer

Respected Editor,

We would like to thank the Editor and Referee for reviewing and providing many useful suggestions to make our manuscript more effective. We revised the manuscript and tried our best by incorporating the suggestions of reviewers properly. In this response, we are presenting a point-by-point summary of the reviewer's Points.

Responses to reviewer 3.

Major concerns:

  1. In table 1 the authors should clarify in the expression of consistency whether the range has been added or whether it is the standard deviation or the standard error.

Response: Thank you for your valuable Point. We have described the range of Consistency in the table 1. The parameters in the table indicate the plastic limit and liquid limit of the soil, as well as the respective standard deviations.

Reviewer 4 Report

The paper addresses an important topic, it is well described, but the description is specific to a book chapter and less so to a scientific paper.

I believe that the paper needs major improvements to be suitable for publication:

- the abstract must contain more information from the research hypothesis, objectives and research methodology;

- current references can be included in the introduction chapter to support the research hypothesis;

- for the formulas used, references must be presented if they are not original;

- the tables and especially the figures must be improved and presented accurately (both the text in them and the format);

- the obtained results must be discussed, related to similar researches and the novelty of the presented researches must be highlighted.

Minor editing of English language required.

Author Response

File Name: Review Report of Research Article sensors

Tittle:     Optmized design of touching parts of soil disinfection machine based on strain sensing and discrete element simulation

Subject:        Authors response to the Editor and Reviewer

Respected Editor,

We would like to thank the Editor and Referee for reviewing and providing many useful suggestions to make our manuscript more effective. We revised the manuscript and tried our best by incorporating the suggestions of reviewers properly. In this response, we are presenting a point-by-point summary of the reviewer's Points.

Responses to reviewer 4.

Major concerns:

  1. The abstract must contain more information from the research hypothesis, objectives and research methodology;

Response: Thank you for your valuable Point. We have reworked the content of the abstract and added a description of the research methods.

  1. Current references can be included in the introduction chapter to support the research hypothesis;

Response: We have included references to vertical spiral blade roll in the introduction to describe their application in agroforestry engineering to support the research hypothesis of this research.

  1. For the formulas used, references must be presented if they are not original;

Response: We rechecked the equations used in the paper and added the reference to equation (2) and equation (3).

  1. The tables and especially the figures must be improved and presented accurately (both the text in them and the format);

Response: We have re-checked the format of the table figures and text throughout the paper and made improvements.

  1. The obtained results must be discussed, related to similar researches and the novelty of the presented researches must be highlighted.

Response: We have added a discussion section before the conclusion of the paper to discuss the similarities and differences with similar researches and to elaborate the advantages of this research. The following are the details of the discussion section:

In this research, we aimed to develop a soil disinfection machine and test its rationality by the machine's tillage performance and disinfection effect in the field. The results of simulation and field trial showed that the machine could achieve a tillage depth stability factor of 94.50% and a disinfection uniformity factor of 85.57%. Compared with the vertical spiral blade roll used in this research, Zhang and Liu et al. used multiple pairs of square knives as rototiller rolls to tillage the soil, and their tillage depth stability reached 95.24% and 94.84%, respectively, which were close to the results of this research. However, since this type of blade roll throws soil in all directions, which can affect the mixing of disinfection chemicals with soil, and their disinfection uniformity coefficient is 64.23%, it can be shown that the vertical spiral blade roll is more suitable for vertical tillage as well as soil disinfection.

Round 2

Reviewer 4 Report

The authors improved the paper. I recommend accepting the paper for publication.